# Development of bitopic nanobody-ligand conjugates targeting G protein-coupled receptors and exhibiting logic-gated signaling

**Shivani Sachdev**[☉], **Swarnali Roy**[☉], **Ross W. Cheloha**[ID]*

Laboratory of Bioorganic Chemistry, National Institutes of Diabetes, Digestive, and Kidney Diseases, National Institutes of Health, Bethesda, Maryland, United States of America

[☉] These authors contributed equally to this work.
* ross.cheloha@nih.gov

## Abstract

G protein-coupled receptors (GPCRs) are the largest family of plasma membrane-embedded signaling proteins. These receptors are involved in a wide array of physiological processes, marking them as attractive targets for drug development. Bitopic ligands, which are comprised of a pharmacophore that targets the receptor orthosteric site and a linked moiety that binds to a separate site, have considerable potential for addressing GPCR function. Here, we report the synthesis and evaluation of novel bitopic conjugates consisting of a small molecule pharmacophore that activates the adenosine A2A receptor (A2AR) linked to antibody fragments (nanobodies, Nbs). This approach leverages the high-affinity and specificity binding of Nbs to non orthosteric sites on engineered A2AR variants to provide bitopic Nb-ligand conjugates that stimulate strong and enduring signaling responses. We further demonstrate that such bitopic conjugates can induce activation by spanning two distinct receptor protomers. This property enables the selective targeting of receptor pairs over either individual receptor, as a form of "logic-gated" activity. We showcase the broad applicability of bitopic conjugates in this context by demonstrating their activity in targeting several pairs of co-expressed receptors, including GPCR monomers from different classes. Furthermore, we demonstrate that this dual-targeting strategy initiates signaling responses that diverge from those induced by monovalent ligands. The ability to target receptor pairs using Nb-ligand conjugates offers a powerful strategy with potential for cell type-selective signaling and implications for GPCR drug discovery efforts more broadly.

## Introduction

G protein-coupled receptors (GPCRs) form the largest family of mammalian cell surface proteins and are well-established drug targets, comprising the target of

**Data availability statement:** All relevant data are within the paper and its Supporting information files.

**Funding:** This work was supported by the NIH Intramural Research Program (NIDDK, 1ZIADK075157, and 1ZIADK075184 R.W.C.) and funding from the NIH Director's Innovation Challenge Award (R.W.C.). The funders had no role in study design, data collection and analysis, decision to publish, or preparation of the manuscript. The findings and conclusions presented in this paper are those of the author(s) and do not necessarily reflect the views of the NIH or the U.S. Department of Health and Human Services.

**Competing interests:** The authors have declared that no competing interests exist.

**Abbreviations:** A2AR, A2A adenosine receptor; AUC, area under the curve; BRET, bioluminescence resonance energy transfer; cAMP, cyclic adenosine monophosphate; DBCO, dibenzylcyclooocyne; ELISA, enzyme-linked immunosorbent assay; GFP, green fluorescent protein; GPCRs, G protein-coupled receptors; MHC-I, Class I major histocompatibility complex; Nbs, nanobodies; PEG, polyethylene glycol; PTHR1, parathyroid hormone receptor-1; XAC, xanthine amine congener.

approximately 35% of currently marketed therapeutics [1]. Their widespread distribution and diverse functional properties make GPCRs prime candidates for drug development. GPCRs are activated by diverse ligands, ranging from small molecules to peptides and proteins. GPCR agonists bind to the orthosteric site of the receptor. This binding event induces a conformational change which, in turn, activates heterotrimeric G proteins and downstream intracellular signaling cascades. Most GPCR drug discovery efforts to date have focused on targeting orthosteric sites. Such sites are highly conserved across GPCR subtypes, which presents challenges for the identification of highly specific ligands. Binding sites outside of the receptor orthosteric pocket, also known as allosteric binding sites, may exhibit more divergence between GPCR subtypes and offer better prospects for the generation of highly specific binders [2]. However, the engagement of allosteric sites frequently fails to induce desired signaling responses. One approach to address this issue focuses on bitopic ligands—molecules comprised of both an orthosteric and an allosteric pharmacophore that are chemically linked to allow targeting of two binding sites [3]. Despite their potential for achieving selectivity, the rational design of bitopic compounds is challenging, especially for class A GPCRs that often lack highly defined secondary binding sites.

Class A, or rhodopsin family GPCRs, represent the largest GPCR subset and have been studied extensively. Most class A GPCRs are characterized by a small fragment exposed to the extracellular space, although many exemptions are known [4]. One representative member, the A2A adenosine receptor (A2AR), is an important target for drug development. Efforts in structure-based molecular docking, rational drug design, medicinal chemistry campaigns, and high-throughput screening have expanded the repertoire of A2AR ligands [5]. These efforts have yielded new compounds with diverse pharmacological profiles; however, there remains a critical need for new approaches to provide ligands with requisite specificity.

Challenges in specificity are further accentuated by the expression of individual GPCRs, such as A2AR, in different tissues. Activation of a GPCR in one tissue can lead to a different physiological response than observed when the same GPCR is activated in a separate tissue or cell type. A2AR is expressed widely in tissues such as the brain, heart, stomach, bladder, liver, immune cells, and adipose tissue [6]. Disease-related changes in GPCR function may occur only in a specific sub-population of cells. Efforts to modulate the function of these widely expressed receptors for therapeutic purposes may be complicated by drug action in other tissues resulting in on-target, off-tissue adverse side effects. One approach to address such complications entails the development of ligands that act in a tissue-specific manner [7]. Ligands that exert activity only upon expression of two distinct targets, analogous to the action of an "AND" logic gate, offer the prospect of inducing biological responses only on cells in which both targets are expressed. Logic-gated activity offers the prospect for targeting widely expressed receptors (such as A2AR) only in specified tissues by making ligand activity conditional on the expression of a second marker preferentially expressed at the anatomical site of interest. Despite the appeal of this approach, the design of ligands with logic-gated activity remains challenging [8,9].

Antibodies and related biologics present unique possibilities for GPCR drug discovery owing to their high affinity and ability to bind receptor surfaces poorly addressed with small molecules [10,11]. Nanobodies (Nbs), derived from heavy chain-only antibodies found in camelids, offer advantages over other modalities due to their small size (12–15 kDa), modularity enabling multimerization, and high antigen-binding affinity [11–13]. Such single-domain antibody fragments sometimes exhibit the capacity to bind GPCR epitopes that are poorly addressed with conventional monoclonal antibodies [11]. These properties mark Nbs as ideal building blocks for assembling multispecific constructs. Past work from our group has shown that linking Nbs to bioactive GPCR ligands, which both bind to the same receptor, provides bitopic conjugates that exhibit specialized and useful properties [14–19]. However, these past studies were limited to GPCR targets bound by peptide ligands. The capacity of these ligands to act in a logic-gated manner has also not been evaluated.

Here, we present a new semi-synthetic, chemical biology approach that provides small molecule-Nb conjugates in which Nb binding tethers linked small molecule agonists near to their site of action to facilitate targeted receptor activation. These engineered bitopic small molecule-Nb conjugates demonstrate high potency activation of A2AR that is fully dependent on the binding of Nb to their epitope target on the cell surface. We further elaborate on this platform to show that Nb-ligand conjugates can act in a logic-gated manner. By linking a GPCR ligand to a Nb that targets a receptor distinct from that bound by ligand, we produce conjugates that exhibit activity only when both targets are co-expressed in a single cell population. Such conjugates do not show activity when either the ligand- or Nb targets are expressed individually, clearly demonstrating logic-gated behavior. This methodology offers a path towards tissue-specific pharmacology, with implications for GPCR drug discovery and therapies with reduced side effects.

## Results

Past work from our group for targeting cell surface receptors focused on the construction of conjugates comprised of Nbs and medium-to-large peptide GPCR ligands [14–19]. In this context, the sites chosen for attachment of linkers to peptide ligands were selected based on structural and structure-activity relationship studies. Linkers were installed at the peptide termini (*N*- or *C*-) distal to the portion of the ligand most intimately engaged with the receptor orthosteric site. Most small-molecule GPCR ligands do not possess such obvious sites for linker installation. To overcome this challenge, we designed small-molecule ligand-Nb conjugates based on the structure of A2AR bound to the adenosine analogue and agonist CGS21680 (CGS) [20]. CGS engages the A2AR orthosteric site while simultaneously projecting a carboxylate group into the extracellular vestibule of the receptor (Fig 1). This feature provided an attractive handle for attachment of a linker to enable construction of CGS conjugates without abrogating ligand agonist properties. Past work has shown that analogues of CGS functionalized at this site retain robust agonist activity [21,22]. We attached an azide group to CGS via a short polyethylene glycol ($PEG_3$) chain (Fig 1 and Fig A in S1 Text). Notably, this approach enables the preparation of bitopic conjugates without modifying the core structure of the CGS pharmacophore.

There are currently no Nbs reported that bind to the extracellular face of A2AR. To facilitate the evaluation of CGS-Nb conjugates that target A2AR, we engrafted tag sequences into the distal N-terminus (extracellular portion) of A2AR. These tag sequences, introduced through engineering of receptor expression plasmids, were comprised of either a Nb that recognizes a 14-mer epitope tag ($Nb_{6E}$ in Fig 1B, right) [18] or tandem epitope tags (BC2, 6E, and ALFA) recognized by other Nbs (Fig 1B, middle). Both constructs contain an ALFA epitope tag [23], recognized the $Nb_{ALFA}$. The engrafted tags serve as high-affinity anchoring sites for either epitope tag-binding Nbs (e.g., **$Nb_{ALFA}$** binds to A2AR-$Nb_{6E}$-**ALFA** or A2AR-BC2-6E-**ALFA**) or an epitope peptide (e.g., **6E peptide** binds to A2AR-$Nb_{6E}$-ALFA). We hypothesized that the Nb-tag interactions would facilitate high-affinity ligand binding, thereby increasing the local concentration of the ligand at the receptor orthosteric site. Each of these Nb-tag pairs have been previously characterized, including in the context of GPCR engineering efforts [14–19]. The affinity of each of these Nb-tag interactions is high, with $K_D$ values that range from to nanomolar (6E tag, BC2 tag), to picomolar (ALFA tag) [18,23–25].

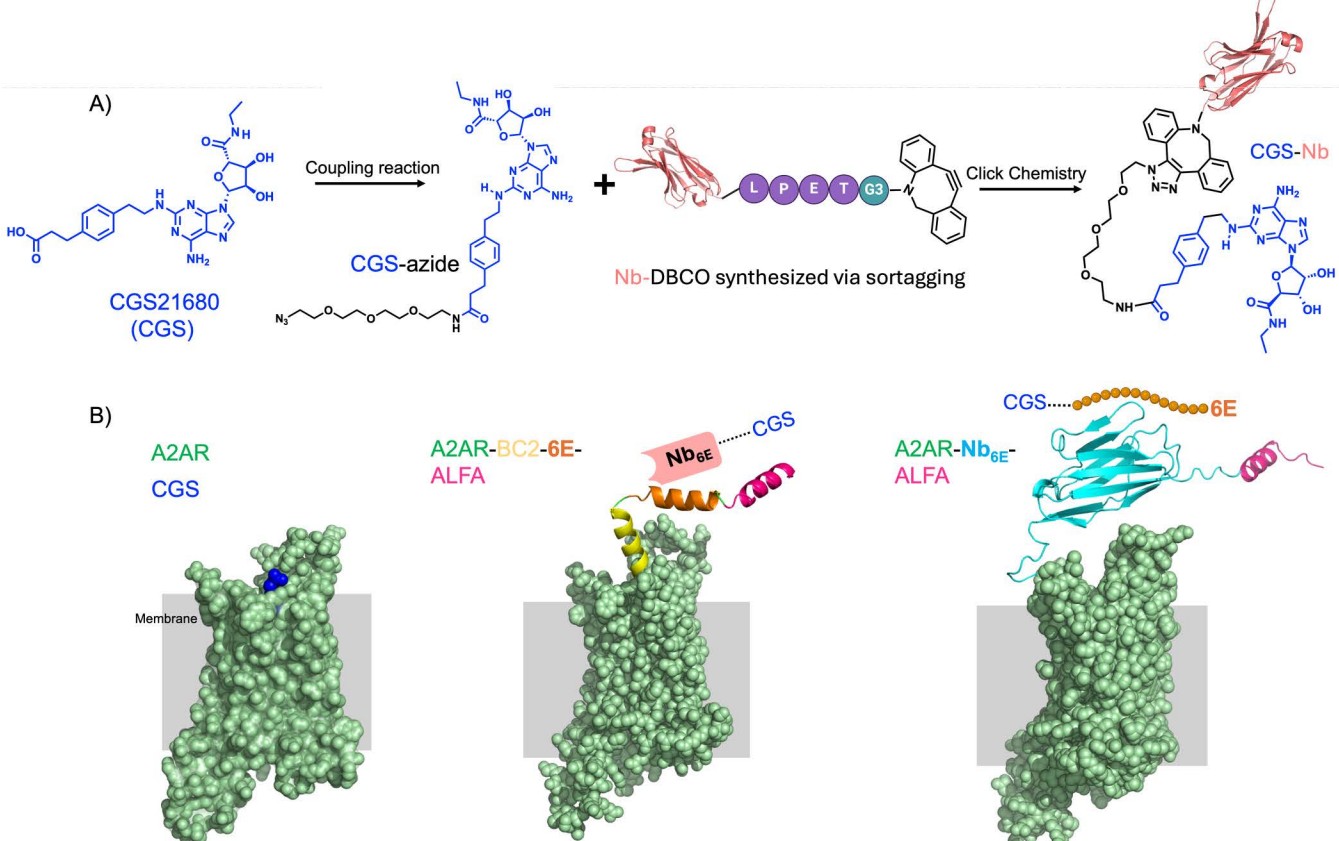

**Fig 1. Composition of engineered bitopic ligands and receptor.** A) Synthetic scheme for production of conjugates comprised of Nbs and a small molecule G protein-coupled receptors ligand (CGS). Detailed synthetic methodology and characterization of small molecules and conjugates is provided in Materials and Methods and Figs A and B in S1 Text. B) Schematic of receptor constructs. (Left) Structure of the unmodified A2AR in complex with CGS (PDB: 4UG2). (Middle) Epitope tags (BC2, 6E, ALFA) were genetically engrafted into the N-terminus (extracellular portion) of A2AR to enable Nb recognition. (Right) The ALFA epitope tag and $Nb_{6E}$ were genetically engrafted into the N-terminus of A2AR. Models of A2AR receptor variants were generated using Alphafold2 via Colabfold (see Materials and methods).

We generated chimeric ligands that bind to both the receptor orthosteric site and a high affinity non-orthosteric (tag) binding site; a method previously established in our laboratory [14–19]. Nbs ($Nb_{6E}$, $Nb_{ALFA}$, and the negative control $Nb_{GFP}$) [26], equipped at their *C*-termini with a sortase A recognition motif (LPETG), were expressed recombinantly in *Escherichia coli*, (Fig A in S1 Text). The Nb *C*-termini were site-specifically modified with a triglycine functionalized dibenzylcyclooctyne (DBCO) probe using sortagging [27]. An azide-labeled analogue of CGS (CGS-azide) and DBCO-labeled Nbs were conjugated through a strain-promoted azide-alkyne cycloaddition ("click") reaction (Fig 1 and Fig A in S1 Text). We also synthesized a set of peptide-based conjugates in which an analogue of CGS was linked to 6E epitope tag peptide, based on previously published methods (Fig B in S1 Text) [18]. The identity of each of the bitopic CGS-Nb and CGS-peptide conjugates were confirmed by analysis with mass spectrometry (Tables A and B in S1 Text). An assessment of CGS-$Nb_{6E}$ for binding to its epitope (6E tag) using an enzyme-linked immunosorbent assay (ELISA) revealed that it retained high-affinity target binding, with affinity very similar to $Nb_{6E}$ (Fig C in S1 Text). This observation suggests that the style of ligand linkage used in this study does not negatively impact the affinity of Nbs for their epitopes.

We assessed the functionality of engineered receptors and the capability of semi-synthetic conjugates to induce their activation by measuring production of intracellular cyclic adenosine monophosphate (cAMP), a common readout

of GPCR-mediated $G\alpha_s$ signaling. Our assay employed HEK293-based cell lines that stably express individual A2AR receptor variants of interest along with a luciferase-based biosensor for real-time monitoring of receptor signaling and cAMP production [28,29]. Each of these cell lines exhibited staining with a fluorophore conjugated analogue of CGS (CGS-AZ647), with A2AR-BC2-6E-ALFA, or A2AR-Nb6E-ALFA showing the highest levels of staining (Fig D in S1 Text). We compared the agonist properties of CGS with CGS-6E, CGS-Nb$_{6E}$, CGS-Nb$_{ALFA}$, and CGS-Nb$_{GFP}$ in cell lines express-ing A2AR, A2AR-BC2-6E-ALFA, or A2AR-Nb$_{6E}$-ALFA. Treatment with CGS demonstrated a concentration-dependent induction of cAMP production across all the receptor variants, validating the functionality of the engineered receptors (Fig 2). Application of engineered Nb- and peptide-CGS conjugates to wild-type A2AR, which does not possess epitope tags, resulted in little activity. This finding aligns with previous work showing that the linkage of Nbs that do not bind to the receptor of interest to ligands with modest affinity drastically diminishes agonist activity [14,17]. Exposure of cells express-ing A2AR-BC2-6E-ALFA to specific CGS-Nb conjugates (CGS-Nb$_{6E}$ and CGS-Nb$_{ALFA}$) resulted in full activation of cAMP production with potencies 25-fold greater than CGS itself (Fig 2B). The conjugation of CGS to negative control Nb (CGS-Nb$_{GFP}$) [26], which recognizes an epitope (green fluorescent protein, GFP) not present, rendered the conjugate essentially inactive at this receptor (Fig 2).

Similar experiments were performed on the cells expressing A2AR-Nb$_{6E}$-ALFA [18]. Here, CGS-6E and CGS-Nb$_{ALFA}$ exhibited a 30-fold enhancement in potency compared to CGS. In contrast, CGS-Nb$_{6E}$ was no longer active since this receptor variant lacks the 6E tag (Fig 2C). We also generated a receptor construct in an enhanced GFP variant was inserted into the receptor ectodomain (A2AR-GFP). The "negative control" CGS-Nb$_{GFP}$ conjugate was more effective than any other ligand tested on this receptor, in contrast to its inactivity on all other A2AR variants lacking GFP (Fig E in S1 Text). These findings confirm that conjugate activity is dependent on Nb-epitope interactions, potentially operating through a ligand tethering mechanism.

The duration of ligand-induced signaling responses can contribute to the strength of the overall biological response induced [30]. To determine whether Nb-mediated tethering of small molecule agonists to their site of action affects the duration of signaling, we measured cAMP responses over an extended period following extensive washout. Briefly, cell lines expressing receptors of interest were stimulated for a defined period, during which peak cAMP production was reached, after which the agonist was washed out. The duration of the CGS-induced cAMP response following washout was relatively short, which is consistent with previous observations (Fig 2) [18]. In contrast, active CGS-Nb conjugates induced cAMP responses that were prolonged compared to CGS (Fig 2). We quantified signaling duration by measuring the area under the curve (AUC) after washout, which revealed substantial differences in the duration of action among the conjugates and the unconjugated ligand (Fig F in S1 Text). This trend suggests that the affinity of the Nb for the tag may promote continued signaling after washout.

Alternative tools and assay formats were enlisted to further explore the activity of bitopic conjugates. A bioluminescence resonance energy transfer (BRET)-based G protein activation assay platform was adapted to measure A2AR activation [31]. This method enables the assessment of ligand activity by measuring the translocation of G protein biosensors, a proximal event in GPCR activation, which minimizes the signal amplification effects that are inherent in methods that measure the production of downstream second messengers (such as cAMP). This BRET assay was applied in the same A2AR-Nb$_{6E}$-ALFA cell line that was used for cAMP-based experiments above. Using this BRET assay, CGS exhibited simi-lar potency to previously reported values [31] and the two conjugates tested (CGS-6E and CGS-Nb$_{ALFA}$) were substantially more potent, in agreement with cAMP assays (Fig G in S1 Text). Ligand-induced changes in BRET signals persisted for the full length of the assay (30 m), indicating enduring stimulation of G protein activation (Fig G in S1 Text). Future studies will benefit from extension of BRET biosensor assays such as TRUPATH or bystander BRET to report on the dynamics of G protein activation following ligand washout [31,32].

Another common complication in assessing GPCR agonists relates to the use of cell lines transfected with exogenous constructs to drive high levels of receptor expression. Such cell lines provide high signal-to-noise assay readouts, but they

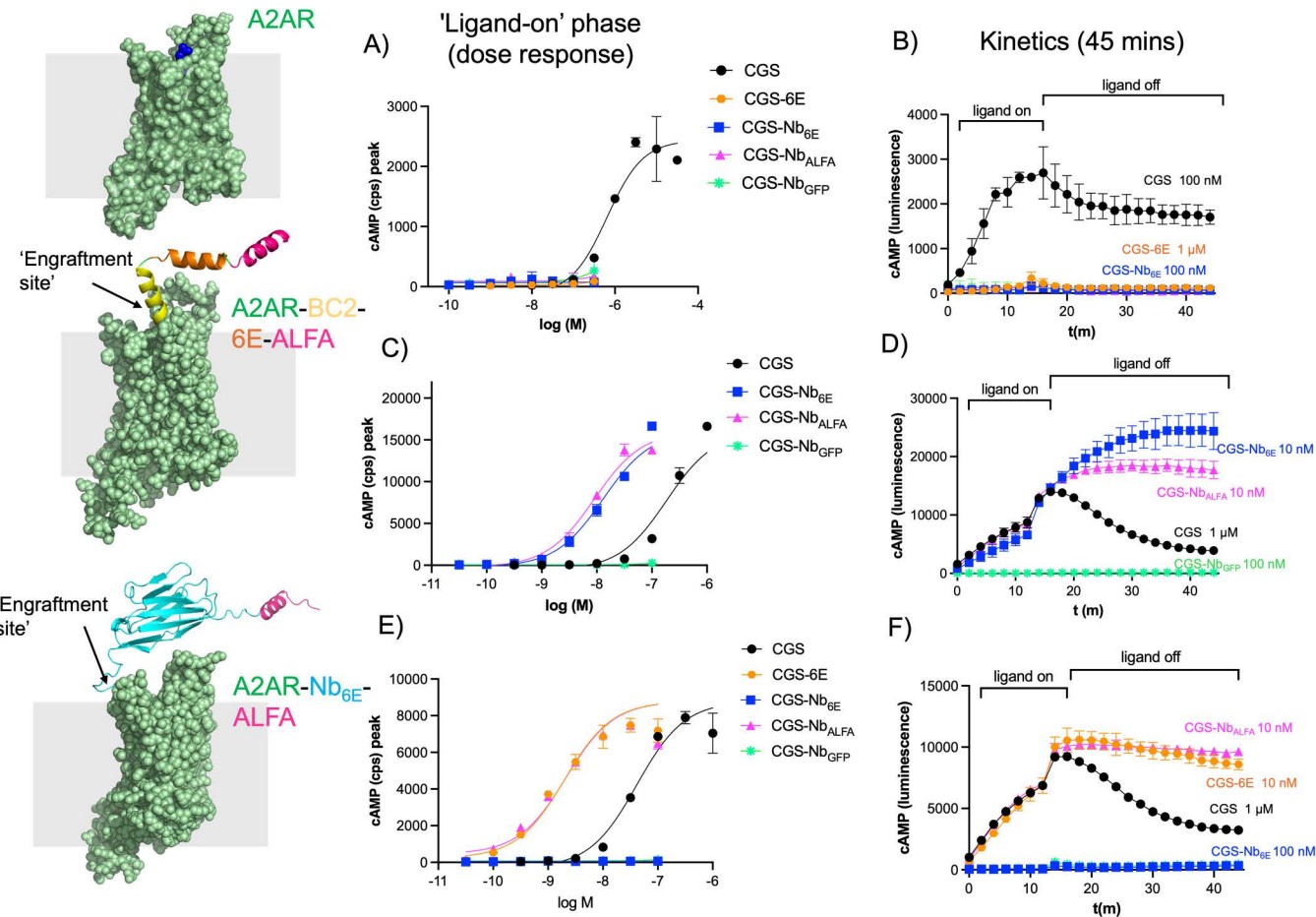

**Fig 2. Assessment of engineered ligand signaling activity.** The activity of Nb-ligand conjugates for stimulating $G_{\alpha s}$-mediated cAMP responses were assessed in cells stably expressing (A, B) A2AR, (C, D) A2AR-BC2-6E-ALFA, or (E, F) A2AR-Nb$_{6E}$-ALFA. (A, C, E) Representative concentration–response curves for ligand-induced cAMP production. Y-axes refer to the peak cAMP response signals recorded approximately 12 min after ligand addition. Responses are quantified by luminescence counts per second (cps). (B, D, F) Representative kinetic measurements of ligand-induced cAMP responses upon ligand addition and after ligand washout. Following ligand addition (12 m, "ligand on"), excess/unbound ligand was removed, fresh media was added to cells, and responses were measured for an additional 30 min ("ligand off"). Compiled and quantified data describing concentration-dependent signaling during the washout period (area under the curve measurements, AUC) is shown in Fig F in S1 Text. Data correspond to mean ± S.D. from an individual experiment with two technical replicates. Concentration–response curves were generated using a three-parameter logistic sigmoidal model. Tabulated data that incorporates 3-5 biological replicates is shown in Table 1. Underlying data can be found in the S1 Data.

sometimes fail to predict ligand action in more physiological contexts. To probe this variable, we generated a cell line that expresses lower levels of A2AR-Nb$_{6E}$-ALFA using limiting dilution and clonal selection. Expression levels of the two A2AR-Nb$_{6E}$-ALFA cell lines were assessed using a flow cytometry-based assay (Fig H in S1 Text) [18]. The high-density A2AR-Nb$_{6E}$-ALFA cell line showed intense staining with the tracer peptide, whereas cells with low receptor density showed substantially (~200-fold) lower staining intensity. This prompted us to assess agonist activity in cells with lower receptor expression. Among all the conjugates tested, CGS-Nb$_{ALFA}$ activated the receptor with the highest potency and efficacy, whereas the natural agonist CGS and CGS-6E only showed activity at high concentrations (Fig H in S1 Text). This observation suggests that Nb-small molecule ligand conjugates are promising candidates for applications in which receptor levels are low, as commonly observed *in vivo*.

We also sought to test the extent to which the engineered bitopic ligands interact with the receptor through a two-site (tag and orthosteric binding) mechanism (Fig 3A). To evaluate the importance of continued engagement at either the

**Table 1. Summary of the activity of ligands for inducing cAMP responses at A2AR and variants.**

| EC$_{50}$ (±SEM), nM Ligands | peak cAMP | Washout AUC |
|---|---|---|
| **A2AR-BC2-6E-ALFA** | | |
| CGS | 144 (25) | 714 (180) |
| CGS-Nb$_{6E}$ | 14 (4) | 7 (1.4) |
| CGS-Nb$_{ALFA}$ | 4.8 (1.3) | 5 (1) |
| CGS-Nb$_{GFP}$ | >10,000 | >10,000 |
| **A2AR-Nb$_{6E}$-ALFA** | | |
| CGS | 52 (8) | 95 (6.7) |
| CGS-6E | 1.8 (0.5) | 0.8 (0.2) |
| CGS-Nb$_{6E}$ | >10,000 | >10,000 |
| CGS-Nb$_{ALFA}$ | 2.6 (0.7) | 0.9 (0.3) |
| CGS-Nb$_{GFP}$ | >10,000 | >10,000 |

EC$_{50}$ values correspond to mean measurements from five independent experiments. Entries denoted ">10,000" indicate that EC$_{50}$ values could not be determined due to weak ligand activity in that assay. "Peak cAMP" EC$_{50}$ values are derived from fitting a sigmoidal concentration–response model to peak luminescence responses recorded after ligand addition (typically measured at 12 m). "Washout" data are derived from fitting a sigmoidal concentration–response model to integrated area under the curve (AUC) values from sequential measurements performed every 2 m for a total of 30 m following ligand removal. $E_{max}$ values are similar across active compounds and are not listed for clarity. Representative concentration–response graphs are shown in Fig 2 and Figs E and F in S1 Text. A comprehensive tabulation of data, which incorporates data shown here, is provided in Table C in S1 Text.

receptor orthosteric site or the Nb-epitope binding site, we introduced a competitor at the initiation of the washout phase (Fig 3B). We added free 6E peptide to compete for the interaction between CGS-Nb$_{6E}$ and A2AR-BC2-6E-ALFA (Fig 3C). Addition of the 6E competitor peptide caused a rapid abatement of the cAMP signaling for CGS-Nb$_{6E}$, which was quantified via AUC measurements (Fig I in S1 Text). In contrast, neither CGS nor CGS-Nb$_{ALFA}$ showed any reduction in signaling when tested under identical conditions, as predicted. The very rapid loss in signaling for CGS-Nb$_{6E}$ upon the addition of 6E peptide suggests that the bitopic ligand does not maintain continuous engagement with the 6E tag binding through the duration of the signaling response.

This observation led us to ask whether engineered bitopic ligands are also susceptible to competitors that target the A2AR orthosteric site. We evaluated the effects of xanthine amine congener (XAC), a potent A2AR receptor orthosteric antagonist, on ligand washout [33]. Upon addition of XAC during the washout phase, both CGS-Nb$_{6E}$ and CGS-Nb$_{ALFA}$ exhibit a modest reduction in signaling duration, also observed in dose-response plots based on AUC measurements (Fig 3D and Fig I in S1 Text). A similar, but less pronounced reduction in signaling was observed upon addition of XAC during the washout phase for CGS. The relatively rapid loss of signaling observed for CGS-Nb$_{6E}$ and CGS-Nb$_{ALFA}$ suggests that these bitopic conjugates may not continuously engage the receptor orthosteric site and may instead toggle between binding the Nb epitope and the receptor orthosteric site. Similar behavior was previously observed in studies of bitopic Nb-ligand conjugates targeting parathyroid hormone receptor-1 (PTHR1) [17].

Success in developing bitopic ligands targeting a single receptor led us to investigate whether this platform could be adapted to address two distinct cell surface proteins (heteromeric conjugates). We envisioned a scenario in which the Nb from a conjugate would bind to one target while the linked ligand would activate a GPCR co-expressed on the same cell. We hypothesized that the activity of heteromeric conjugates would depend on cellular expression of both the subject of Nb binding and the GPCR targeted by ligand. Such dual dependency would enable development of conjugates that exhibit "AND" logic-gated behavior in which biological responses are only induced in cells co-expressing both targets.

 

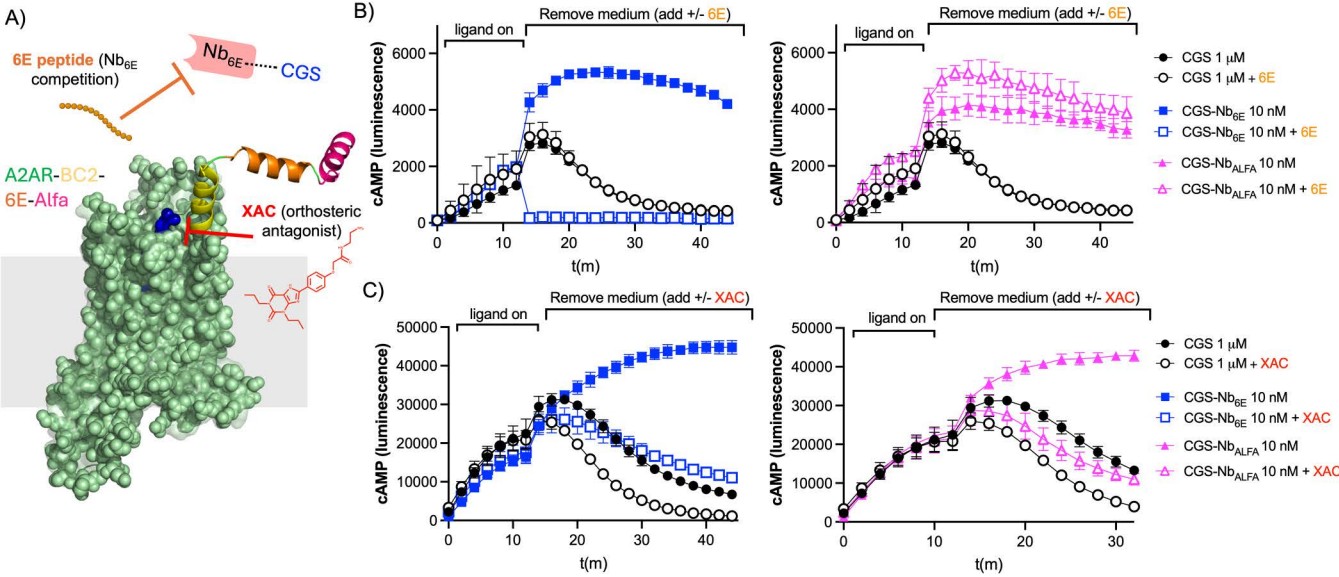

**Fig 3. Mechanistic studies of washout signaling responses.** A) Schematic of a washout competition assay. Ligand washout assays were performed using cells expressing A2AR-BC2-6E-ALFA as described above with the modification that 6E peptide or xanthine amine congener (XAC) was added during the washout phase. B) Time course for cyclic adenosine monophosphate (cAMP) production in HEK293 cells stably expressing A2AR-BC2-6E-ALFA in the presence or absence of competitor 6E peptide. Note that identical CGS signaling data is shown in the left and right graphs to enable clearer visualization. Quantified concentration–response area under the curve data are provided in Fig I in S1 Text. C) Analogous data for cAMP washout responses in the presence or absence of the antagonist, XAC. Note that identical CGS signaling data is shown in the left and right graphs to enable clearer visualization. Agonist potency parameters derived from three independent experiments are summarized in the Fig I in S1 Text. Underlying data can be found in the S1 Data.

We first applied this approach to target two different GPCRs chosen from different receptor classes. A2AR is a class A GPCR, whereas PTHR1 is a member of class B, which is characterized by a large extracellular domain involved in ligand recognition and receptor activation [34]. We built upon the approach described above to synthesize a new set of conjugates based on linkage of a previously characterized PTHR1-binding Nb (Nb$_{PTHR1}$) [14,17] with CGS. The biological activity of the CGS-Nb$_{PTHR1}$ conjugate was assessed in an assay with cells expressing PTHR1 and/or A2AR. As expected, CGS-Nb$_{PTHR1}$ was inactive on cells expressing either A2AR or PTHR1 alone (Fig 4A and 4B). In contrast, co-expression of both receptors led to robust bioactivity for CGS-Nb$_{PTHR1}$ (Fig 4C). We hypothesized that variation in the levels of PTHR1 and A2AR, both in absolute terms and relative to each other, might lead to variation in responses to the heteromeric conjugates. To address this possibility, we evaluated CGS-Nb$_{PTHR1}$ in cells with varying receptor expression levels, induced by plasmid transfection (Fig 4D—4G). CGS-Nb$_{PTHR1}$ induced robust cAMP responses when equivalent amounts of plasmid were used for cell transfection (Fig 4C). Ratios of plasmids of up to 10:1 (w/w basis) were also tested (Fig 4H and Table D in S1 Text). Using a 10:1 ratio of A2AR:PTHR1-encoding plasmids, CGS-Nb$_{PTHR1}$ was still effective in stimulating a cAMP response, albeit with somewhat lower efficacy compared to CGS (Fig 4G). Conversely, when plasmid encoding PTHR1 was used at 10-fold higher levels than the plasmid for A2AR, CGS-Nb$_{PTHR1}$ demonstrated cAMP-inducing potency that exceeded the response seen with CGS alone (Fig 4D). These observations suggest that the activity of Nb-ligand conjugates correlate with levels of Nb-binding target expression. Further, a high level of expression of the Nb-target can enable heteromeric Nb-ligand conjugates to exhibit biological activity that exceeds the response seen with same ligand alone in an identical context. These findings illustrate the principle of receptor co-expression as a means of restricting heteromeric Nb-ligand conjugate activity (Fig 4I).

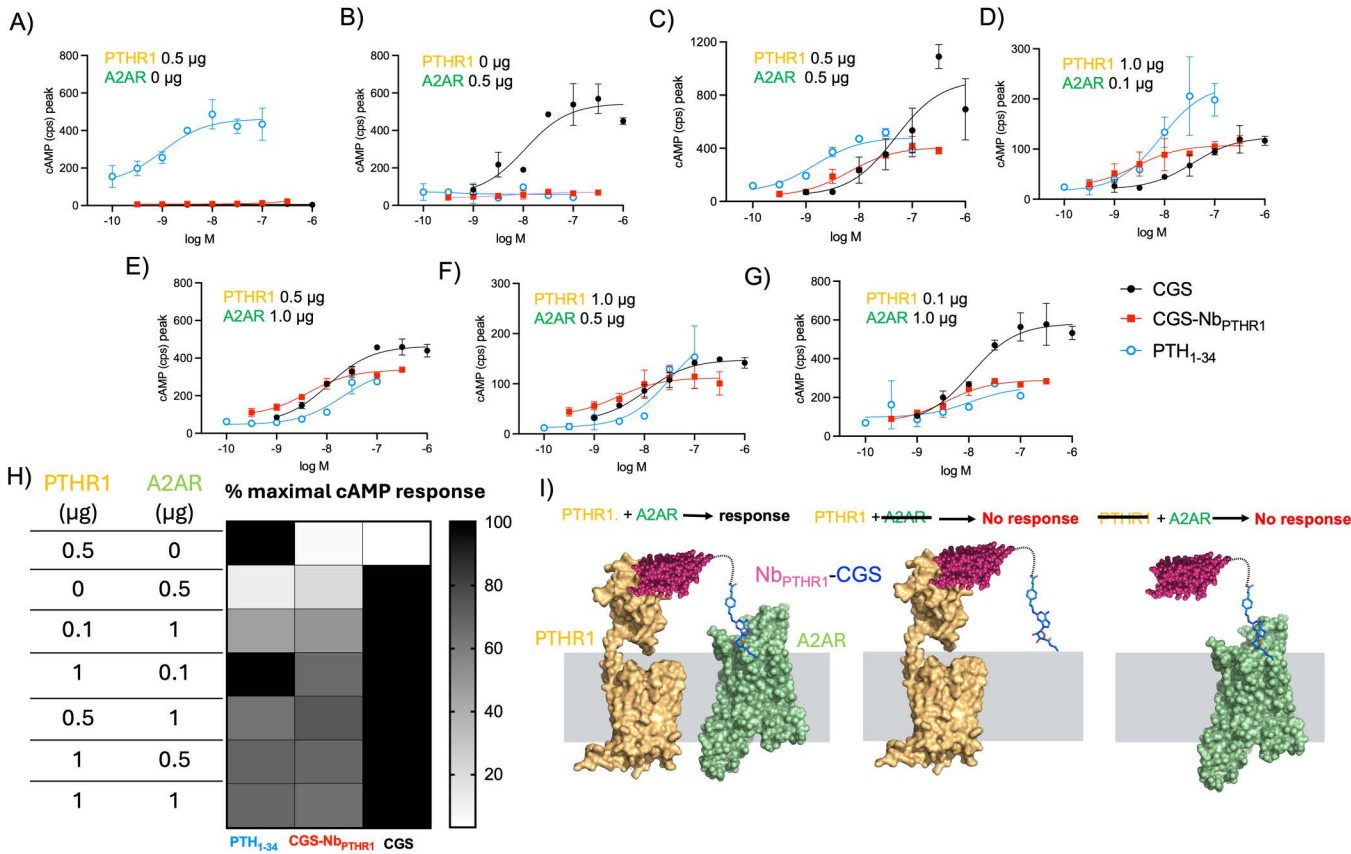

**Fig 4. Logic-gated activation of signaling by targeting G protein-coupled receptor pairs using an engineered bitopic ligand. A–G)** Representative concentration–response curves for induction of cyclic adenosine monophosphate (cAMP) responses in cells co-transfected with varying amounts of plasmids encoding parathyroid hormone receptor-1 (PTHR1) and/or A2A adenosine receptor (A2AR). Data in panels A–G correspond to mean±SD from technical duplicates in a single representative experiment. H) Heat map depicting the maximal cAMP responses induced by the ligands upon transfection of cells with varying amounts of A2AR and PTHR1 encoding plasmids. Corresponding numerical data are provided in the Table D in S1 Text. I) Proposed mechanism of receptor-dependent activity of bitopic ligands. Structures of PTHR1 and A2AR used in this panel were generated using Alphafold2 (see Materials and methods). Underlying data can be found in the S1 Data.

To assess the adaptability of the heteromeric Nb-ligand conjugate-based approach, we also measured the activity of CGS-Nb$_{PTHR1}$ in a HEK293 cell line stably expressing PTHR1 (HEK-PTHR1, instead of transient transfection as above) [29] that was subsequently transiently transfected with A2AR-Nb$_{6E}$-ALFA (Fig J in S1 Text). Unconjugated ligands (PTH$_{1–34}$, PTH$_{1–11}$, and CGS) were included as controls to validate receptor expression and functionality. Consistent with results above, CGS-Nb$_{PTHR1}$ induced a concentration-dependent increase in cAMP production in cells co-expressing PTHR1 and A2AR-Nb$_{6E}$-ALFA. This conjugate was inactive in cells not expressing the A2AR variant (Fig J in S1 Text). Past work with Nb-ligand conjugates targeting PTHR1 revealed highly selective activation of cAMP signaling without concomitant induction of β-arrestin recruitment (biased agonism) [17]. Other work has shown that unlike most GPCRs, activation of A2AR does not stimulate arrestin recruitment [31]. Hence, β-arrestin recruitment assays were not performed for conjugates targeting this pair of receptors.

We also tested a different configuration of heteromeric Nb-ligand conjugates in which the directionality of Nb-binding and ligand targeting moieties was reversed. Here, a weakly active ligand for PTHR1 (PTH$_{1–11}$) was linked to Nb$_{ALFA}$. The choice of PTH$_{1–11}$ was based on past precedent with Nb-ligand conjugates [14]. The PTH$_{1–11}$-Nb$_{ALFA}$ conjugate exhibited a

20-fold increase in cAMP-induction potency relative to $PTH_{1-11}$ in cells co-expressing PTHR1 and $A2AR\text{-}Nb_{6E}\text{-}ALFA$ (Fig J in S1 Text). In contrast, cells lacking $A2AR\text{-}Nb_{6E}\text{-}ALFA$ were nearly unresponsive to $PTH_{1-11}\text{-}Nb_{ALFA}$, demonstrating the requirement for co-expression of both targets for signaling. These findings confirm the activity of heteromeric CGS-Nb conjugates under a variety of conditions.

Although past work had shown that Nb-ligand conjugates in which both Nb and ligand targeted PTHR1 were unable to induce receptor internalization, it was unclear whether Nb-ligand conjugates co-targeting A2AR and PTHR1 might behave differently. To probe this question, we performed a cell-based ELISA to measure the disappearance of receptor from the cell surface upon treatment with agonists (Fig K in S1 Text). Only the prototypical PTHR1 agonist $PTH_{1-34}$ stimulated a decrease in the level of cell surface PTHR1. In contrast, neither $PTH_{1-11}\text{-}Nb_{ALFA}$ nor $CGS\text{-}Nb_{PTHR1}$ reduced the level of PTHR1 detected at the cell surface. This observation may be connected to the previous finding that $PTH_{1-11}$-Nb conjugates failed to induce β-arrestin recruitment [17]. It is also relevant that A2AR activation does not stimulate substantial recruitment of β-arrestin or receptor internalization [31,35]. Previous work demonstrating that bivalent conjugates targeting CXCR7 (a GPCR) can induce target internalization [36] suggest that further studies will be beneficial in revealing contexts in which bivalent conjugates stimulate target downregulation.

Further variations upon the heteromeric Nb-ligand conjugate strategy were also tested. We constructed a conjugate in which CGS is linked to a Nb that binds to a cell surface protein complex, Class I major histocompatibility complex (MHC-I) [37], which is naturally expressed at high levels on most cells. Application of a $CGS\text{-}Nb_{MHC\text{-}I}$ conjugate to cells expressing an A2AR variant resulted in negligible cAMP production (Fig L in S1 Text). We confirmed high levels of MHC-I expression and Nb binding to HEK cells using flow cytometry (Fig L in S1 Text). These findings demonstrate that high-level expression of a Nb-bound target is not sufficient to ensure the activity of Nb-ligand conjugates.

Whether the composition and length of the linker connecting Nb to ligands in heteromeric conjugates was important for biological activity was also tested. Our initial choice of linker length was based on previous studies from our group using CGS-peptide conjugates, in which a medium-length PEG linker was effective [19]. CGS-Nb conjugates with varying linkers were synthesized using the same synthetic approach described above (Fig M in S1 Text). We hypothesized that longer linkers might alleviate distance constraints that could have a negative impact on conjugate activity. Linkers were incorporated with linear chains of polyethylene glycol (PEG) building blocks either 4- or 24-units in length. These conjugates were tested in cells co-expressing PTHR1 and A2AR. This collection of $CGS\text{-}Nb_{PTHR1}$ conjugates showed similar activities for inducing cAMP responses (Fig M in S1 Text). The conjugate containing the longest linker, $CGS\text{-}(PEG)_{24}\text{-}Nb_{PTHR1}$, demonstrated a statistically significant reduction in maximal response compared to the prototype $CGS\text{-}Nb_{PTHR1}$ ligand conjugate (Fig M in S1 Text). This reduction in efficacy may be related to longer linkers failing to preserve the enhancements in local concentration enabled by shorter linkers.

We also wondered whether the Nb-ligand conjugate could induce signaling by bridging two receptors expressed in distinct cell populations. We co-cultured HEK293 cells expressing either PTHR1 (Nb target) or cells expressing A2AR (ligand receptor) (Fig N in S1 Text). The $CGS\text{-}Nb_{PTHR1}$ conjugate failed to induce intercellular activation of GPCR signaling, suggesting that this tethering mechanism functions only when a single cell population co-expresses both receptors.

The generalizability of receptor activation dependent upon coexpression of two receptors was assessed in a separate system. In this new version, A2AR was coexpressed with glucagon-like peptide receptor with an N-terminal 6E tag (GLP1R-6E) in HEK293 cells. Like PTHR1, GLP1R is a class B GPCR that is bound by a Nb previously characterized by our group ($Nb_{GLP1R}$) [17]. We compared $CGS\text{-}Nb_{6E}$ and $CGS\text{-}Nb_{GLP1R}$ to the endogenous peptide agonist of GLP1R (GLP-pep, see Table G in S1 Text for sequence) [17]. When tested in cells co-expressing GLP1R-6E and A2AR, $CGS\text{-}Nb_{6E}$ and $CGS\text{-}Nb_{GLP1R}$ induced cAMP responses similar to CGS (Fig 5). In contrast, these conjugates were completely inactive in cells only expressing GLP1R-6E (Fig 5 and Table E in S1 Text), thus illustrating the adaptability of heteromeric Nb-ligand conjugates for logic-gated signaling.

The above examples involve co-targeting two receptors that primarily signal through the $G_{\alpha s}$ pathway, with one receptor coming from class B GPCRs (PTHR1, GLP1R) and the other from class A (A2AR). We wondered whether we could use a similar approach with two class A GPCRs, or receptors that primarily signal through pathways other than $G\alpha_s$ (see below). To assess the first possibility, we produced conjugates comprised of Nbs that target mu-opioid receptor with an N-terminal (extracellular) 6E tag (MOR-6E). We assessed activation via (co-) expression of MOR-6E and A2AR. We first evaluated whether CGS-Nb conjugates could induce cAMP responses when the Nb component bound MOR-6E. CGS-Nb$_{6E}$ was indeed active for inducing cAMP responses in cells co-expressing MOR-6E and A2AR (Fig O in S1 Text), and this activity was absent in cells expressing A2AR alone. In contrast to examples above, the efficacy of the cAMP response induced by CGS-Nb$_{6E}$ was lower than that of CGS alone (Fig O in S1 Text).

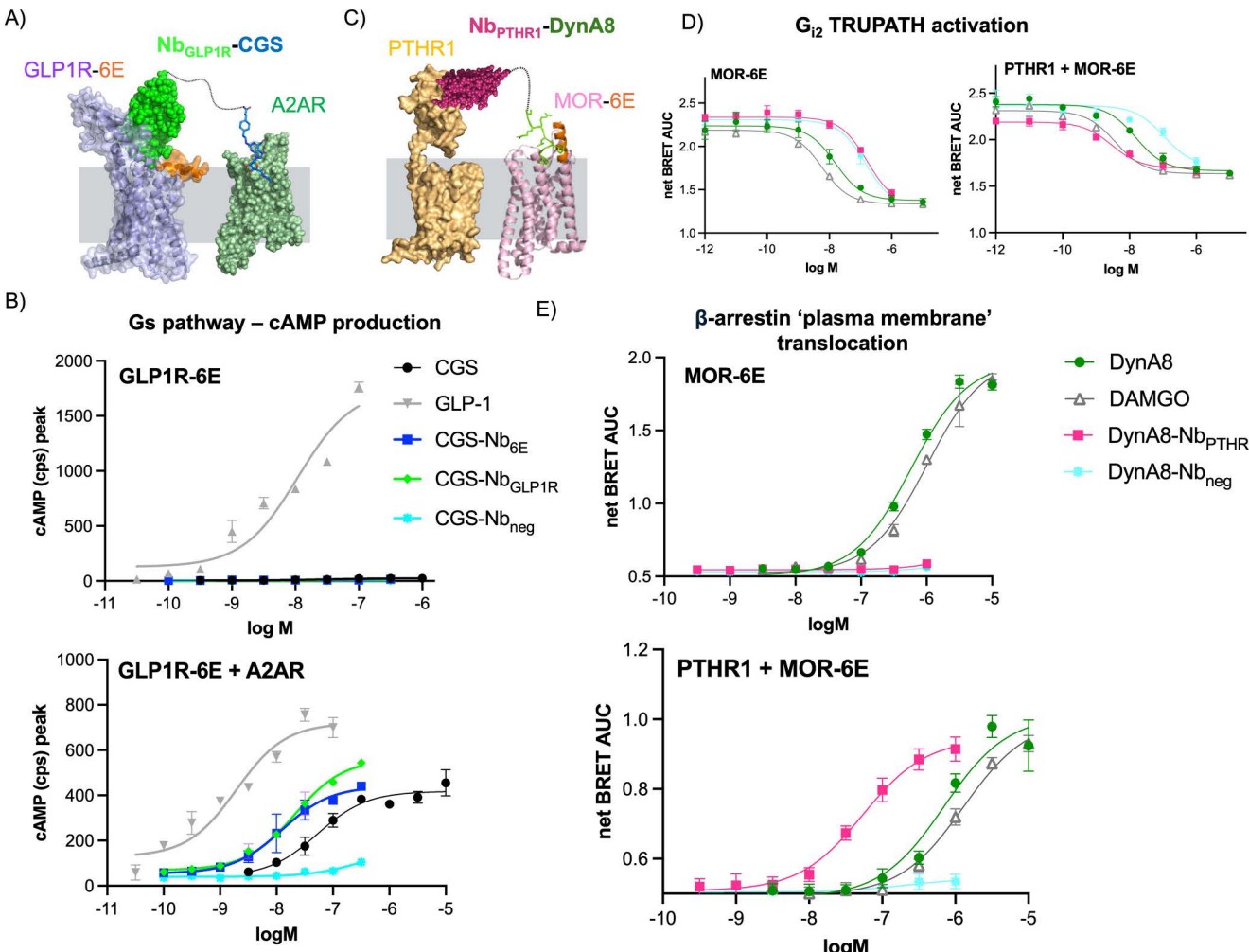

**Fig 5. Extension of bitopic Nb-ligand conjugates to alternative G protein-coupled receptor pairs and signaling pathways.** A) Schematic depicting activation of the GLP1R-A2A adenosine receptor (A2AR) pair by biopic Nb-ligand conjugates. B) Concentration–response assessment of ligand-induced cyclic adenosine monophosphate (cAMP) production in cells expressing GLP1R-6E and/or A2AR. C) Schematic showing activation of MOR-6E and PTHR1 signaling by bitopic conjugates. Note that MOR activation induces cellular responses ($G\alpha i$ activation, β-arrestin recruitment) not mediated by A2AR activation. D) Ligand-induced $G\alpha_{i2}$ activation (TRUPATH biosensor, BRET) and **E)** β-arrestin recruitment (BRET) in cells expressing MOR-6E and/or PTHR1. Data points correspond to mean±SD from technical replicates in a representative experiment. Agonist potency parameters derived from three independent experiments are summarized in the Tables E and F in S1 Text. Underlying data can be found in the S1 Data.

Previous work from our laboratory showed that Nb-ligand conjugates in which both components target PTHR1 strongly activate Gα$_s$ signaling without inducing recruitment of β-arrestin 2, in a dramatic example of pathway-selective signaling [17]. Pathway selective signaling was correlated with the ability of PTH-Nb conjugates to act across two separate receptor protomers. We wondered whether targeting two distinct types of receptors would also result in pathway-selective signaling. To test this, we developed Nb-ligand conjugates that target MOR-6E and PTHR1. Worth noting, activation of either MOR or PTHR1 is known to stimulate the recruitment of β-arrestin, although these receptors signal through opposing G-protein pathways (Gα$_i$ versus Gα$_s$, respectively).

We produced conjugates consisting of Nb$_{PTHR1}$ and a fragment of the MOR agonist peptide Dynorphin A (DynA8-Nb$_{PTHR1}$, see Table G in S1 Text for DynA8 sequence) using a synthetic approach analogous to that used to generate Nb-PTH agonists [17]. DynA8-Nb$_{PTHR1}$ was compared to a prototypical MOR agonist (DAMGO), DynA8 itself, and a conjugate consisting of DynA8 linked to a negative control Nb that binds an epitope absent from the cell surface (DynA8-Nb$_{neg}$). We evaluated DynA8-Nb$_{PTHR1}$ in assays measuring Gα$_{i2}$ activation [32]. DAMGO and DynA8 showed comparable activities in stimulating Gαi in cells expressing MOR-6E alone and in those co-transfected with MOR-6E and PTHR1 (Fig 5 and Table F in S1 Text). In contrast, DynA8-Nb$_{PTHR1}$ was substantially less active in cells expressing MOR-6E alone compared to cells expressing both MOR-6E and PTHR1, where it showed activity like that of DAMGO and DynA8 (Fig 5 and Table F in S1 Text).

DynA8-Nb conjugates were also assessed in assays measuring the recruitment of β-arrestin to the plasma membrane upon GPCR activation [31]. Surprisingly, DynA8-Nb$_{PTHR1}$ exhibited a potency for stimulating β-arrestin 2 recruitment that was >10-fold higher than DAMGO or DynA8 when MOR-6E and PTHR1 were co-expressed (Fig 5). DynA8-Nb$_{PTHR1}$ was completely inactive for stimulating arrestin recruitment in cells expressing MOR-6E alone (Fig 5 and Table F in S1 Text). These findings are in sharp contrast with past findings using bitopic Nb-ligand conjugates that solely target PTHR1, which showed negligible β-arrestin 2 recruitment [17].

To gain further mechanistic insights into the enhanced β-arrestin recruitment shown by DynA8-Nb$_{PTHR1}$ relative to conventional MOR ligands, we performed additional experiments. We varied the levels of plasmids encoding MOR-6E and PTHR1 used to induce receptor expression via transfection. We then measured the activity of DynA8-Nb conjugates in these transfected cells. The use of higher amounts of MOR-6E-encoding plasmid compared to PTHR1 (MOR-6E to PTHR1 ratios 3:1 and 10:1) led to weaker activity for DynA8-Nb$_{PTHR1}$ relative to DAMGO in β-arrestin recruitment assays (Fig P in S1 Text). In contrast, the use of higher levels of PTHR1-encoding plasmid (MOR-6E to PTHR1 ratios 1:3 and 1:10) led to DynA8-Nb$_{PTHR1}$ showing greater activity relative to DAMGO (Fig P in S1 Text). This data demonstrates that high levels of Nb target expression led to high activity for Nb-ligand conjugates. These findings were unexpected and demonstrate for the first time that Nb-ligand conjugates co-targeting two receptors can robustly stimulate signaling through pathways other than Gα$_s$ (Gα$_i$ and β-arrestin), with activities that surpass that of the parent ligand alone.

## Discussion

The bitopic conjugates developed in this study offer two major advances. First, we describe Nb-ligand conjugates made from a small molecule agonist (CGS21680) of A2AR (Fig 1). This represents a major extension from past work in which ligands used for conjugate synthesis were restricted to peptide agonists [14–17,38]. Many GPCRs are activated only by non-peptide small molecules [1], and the approaches described here should be useful for developing Nb-ligand conjugates to target these receptors, as shown for the CGS-Nb conjugates described above. Second, we harnessed the ability of bitopic conjugates to induce activation through spanning two receptor protomers to enable selective targeting of receptor pairs expressed on a single cell (Figs 4 and 5). We showed that such Nb-ligand conjugates only exhibit activity when both the Nb- and ligand targets were co-expressed. Such dual-target dependent signaling offers a promising path for generating conjugates with activity only within a specified biological niche while minimizing on-target, off-tissue side effects [39]. The expression of A2AR across various tissues [40,41] presents opportunities for exploring the consequences

of tissue-restricted activation of a widely expressed receptor. For example, the functional effects of A2AR activation in the brain are likely to differ from those observed upon activation in the immune cells [6]. CGS-Nb conjugates exhibit some of the requisite properties needed for such investigations, although further exploration, particularly in native tissue contexts, is needed.

One notable observation from these studies is that CGS-Nb conjugates induced more enduring signaling responses at engineered A2AR compared to the CGS alone (Figs 2 and 3). This observation suggests that the sustained activation of bitopic conjugates is related to the distinct binding sites of the Nb and CGS, with cooperative interactions leading to prolonged engagement with the receptor. It remains unclear whether the Nb and ligand components from a single conjugate bind simultaneously or if only one component can bind at a time. Studies of CGS-Nb conjugates with longer linkers suggest that simultaneous engagement is feasible but is perhaps not an important contributor to prolonged signaling responses (Fig M in S1 Text). Structural studies of CGS-Nb conjugates bound to receptor will be crucial in addressing these questions. Long-acting ligands have particular significance for A2AR, a well-established therapeutic target for several diseases. Studies on human A2AR have demonstrated a strong correlation between ligand activity *in vivo* and receptor residence time. For example, the strong efficacy of the long-acting A2AR agonist UK-432097 in addressing pulmonary inflammation suggests that extended residence times may enhance therapeutic efficacy [6,42]. The bitopic targeting strategy developed here offers a means for rationally designing ligands with favorable pharmacological properties.

The approach described in this work has parallels with past efforts to develop molecules that selectively target pre-specified GPCR assemblies [43]. GPCR monomers such as A2AR often dimerize or oligomerize with other GPCRs in the membrane when co-expressed in a single cell [44,45], presenting an enticing opportunity to develop highly specific tools for modulating receptor function [46,47]. However, the design of ligands with specificity for GPCR dimers or higher order oligomers remains challenging. We reasoned that simultaneously leveraging both antibody (nanobody) and ligand binding could provide a path toward selective targeting of GPCR heteromers [3]. We have shown that such Nb-ligand conjugates are active only on cells expressing both targets ("AND" logic gate) [8]. While such properties are potentially accessible using traditional ligand design strategies, the modularity of Nb-ligand conjugates may enable more efficient extension to new receptor pairs. *In situ* formation of multivalent conjugates offers another appealing path forward [48].

Emerging evidence highlights the functional roles of GPCR assemblies in mediating biochemical processes that were previously assigned to individual receptors [46]. Studies have shown that GPCR assemblies can consist of receptors that signal either through the same G-protein pathways (e.g., A2AR-β1/β2 adrenergic receptor pairs, via $Gα_s$) or through opposing G-protein pathways (e.g., A2AR-D2R pairs, via $Gα_s$ and $Gα_i$ interactions). A comprehensive understanding of how GPCR assemblies integrate ligand stimuli and environmental cues to generate unique cellular responses remains obscure. Here, we examined signaling induced upon expression of various GPCR pairs, including receptors from different classes and with differing cognate G proteins (such as A2AR-PTHR1, A2AR-GLP1R, A2AR-MOR), when targeted by Nb-ligand conjugates (Fig 5). These experiments revealed several interesting trends. The strongest signaling responses induced by Nb-ligand addition were observed in cells where the Nb-targeted receptor was expressed at higher levels than the ligand-binding receptor. We also showed for the first time that bitopic conjugates can induce signaling through signaling partners besides $Gα_s$ ($Gα_i$, β-arrestin). One surprising observation was that a bitopic conjugate (DynA8-Nb) targeting cells expressing PTHR1 and MOR more robustly induced arrestin recruitment than the ligand (DynA8) alone (Fig 5). These results contrast previous examples in which bitopic ligand conjugates exclusively targeting PTHR1 exhibited strong G-protein activation with minimal arrestin recruitment [17]. These findings pose intriguing questions about the ability of co-expressed GPCR pairs to generate specialized signaling responses, compared to situations in which receptors are expressed individually. These trends are highly relevant in physiological contexts, where single cells co-express a multitude of membrane proteins, potentially expanding the complexity of receptor function in native cellular environments.

The use of Nb-ligand conjugates for logic-gated receptor activation offers several appealing opportunities for future investigation. For example, activation of glucagon family receptors (GLP1R, GIPR, GCGR) shows therapeutic efficacy in

treating type-2 diabetes and obesity [49]; however, these receptors are expressed in multiple tissues [50]. The contribution of tissue-specific pharmacology to overall therapeutic efficacy and side effect profiles is unclear. Logic-gated bitopic conjugates could facilitate cell type-specific receptor activation based on co-expression of a secondary cell marker, thereby enabling mechanistic investigations. The modular and programmable nature of our strategy could enable the possibility of engineering ligands for a diverse array of physiologically relevant GPCRs at their site of action *in vivo*. Extension into new areas of physiology will benefit from new approaches to identify Nbs that bind to GPCRs [11,51–55]. Further work will be needed to design Nb-ligand conjugates that can act by spanning two cells (intercellular function) to act across cell types in heterogeneous tissues.

## Materials and methods

### Materials

Specified reagents were purchased from commercial vendors: PTH1−34 (Genscript, RP01001), DAMGO (Caymen Chemical, #21553), D-luciferin (GoldBio, LUCNA-1G), Coelenterazine prolume purple (Nanolight Technology, #369-1), and CGS21680 (SelleckChem, #S2153), XAC (Cayman Chemical, #23077), and THPTA (Vector Laboratories, #CCT-1010-100). Solvents (dimethylformamide, dichloromethane, acetonitrile, diethyl ether, and dimethylsulfoxide) and other chemical reagents (diisopropylethylamine, diisopropylcarbodiimide, Oxyma, trifluoroacetic acid, triisopropylsilane) were purchased from Sigma Aldrich. Other reagents were synthesized in-house (or purchased) as described in relevant figure captions and in methods.

### Cell culture and analysis

Methods for cell culture, cell transfection, and analysis of ligand binding with flow cytometry are described in S1 Text. Plasmids produced using a commercial gene synthesis and cloning service (Genscript) were used to transfect cells to express GPCRs of interest.

### Measurement of cAMP responses (including washout and competition assays)

Receptor activation via the Gs signaling pathway was quantified using a cAMP luciferase reporter assay [28,29]. HEK293 cells expressing cAMP reporter and receptors of interest were trypsinized and seeded in a white-walled 96-well flat clear-bottom plate (Corning #3610). The cells were grown to full confluency prior to use in bioassays. Prior to bioassays, the growth medium was removed, and serum-free $CO_2$-independent medium containing 0.5 mM D-luciferin was added. Luminescence was monitored using a multiwell plate reader (Biotek Neo2 plate reader) for 10 min to establish a stable baseline. Ligands diluted in $CO_2$-independent medium were added to the wells for a final well volume was 100 µL. The luminescence response was measured every 2 min for 12 min total. The peak luminescence responses (typically measured at 12 m) were used to generate concentration-response curves. The concentration–response curves were fitted to individual experiments utilizing a three-parameter sigmoidal dose-response model (log[agonist] versus response) using GraphPad Prism to produce $EC_{50}$ and $E_{MAX}$ values.

For the measurement of cAMP washout responses, the cells were stimulated with an agonist for a specified period (typically 12 m) as described above (ligand-on phase). At this time, the medium containing ligands was discarded. New $CO_2$ independent medium containing fresh luciferin was added to all wells and luminescence responses were measured every 2 min for an additional 30 min (ligand-off phase). For competition-washout assays, the experimental protocol was performed as described above, except that competitors were added during the ligand-off phase. Concentration–response curves for washout assays were generated by quantifying AUC for kinetic measurements of luminescence for the washout phase. Uncertainty measurements at each time point during the washout phase were propagated to provide standard error values for AUC measurements, which are depicted as error bars in relevant graphs.

## Bioluminescence resonance energy transfer (BRET) assays to measure β-arrestin translocation and G-protein trafficking ($G_i$ TRUPATH and $G_s$ dissociation)

HEK293 cells expressing the receptor of interest were seeded in a 10-cm dish using approximately 2,250,000 cells and allowed to adhere and grow for 12 h at 37°C. Adherent cells were then transfected with plasmids encoding membrane-tethered BRET acceptor plasmid (rGFP-CAAX or rGFP-FYVE, 1,008 ng) and RlucII-conjugated β-arrestin 2 BRET donor plasmid (β-arrestin 2- RlucII, 72 ng) using Lipofectamine 3000 as previously described [17,31]. For $G\alpha_s$ dissociation BRET assays, 1,008 ng of rGFP-CAAX plasmid and 144 ng of $G\alpha_s$-RlucII plasmid were transfected [17,31].

For TRUPATH $G_i$-activation assay, 1 μg of $G_i$ TRUPATH plasmid encoding $G\alpha_{i2}$, $G\beta_3$, and $G\gamma_9$ was used for transfections [32]. Transfected cells were grown for an additional 20–24 h. After transfection, cells were resuspended in standard growth medium and seeded into a white, clear-bottom 96-well plate at a density of 70,000 cells/well. After 24 h of growth, plated cells were incubated in assay buffer consisting of Hanks-buffered saline solution supplemented with 5 mM HEPES and 1.3 μM Prolume Purple for 5 m before the addition of serial dilutions of ligand or nanobody-ligand conjugates. BRET signals were measured on a Biotek Neo2 plate reader with measurement of emissions at 400 and 510 nm using appropriate filters. Plate reads took approximately 150 sec and were repeated for a total of 30–40 min. BRET ratios (515/410 nm) were calculated for each well at each time point. Concentration–response curves were generated by quantifying the AUC of kinetic measurements of BRET ratios using GraphPad Prism. Standard error values are calculated by propagating the standard deviations measured at each time point.

## Alphafold-enabled receptor structural modeling

The online tool ColabFold [56] implementing Alphafold2 was used to predict tagged receptor and Nb-receptor complexes [57]. The default AlphaFold2 settings were applied, resulting in generation of five models for each input sequence. Top-ranking models were applied to generate all graphics using PyMOL. In certain cases, models were aligned with related experimentally determined structures using the "align" command in PyMOL.

## Chemical synthesis

A description of the synthesis, purification, and analysis of peptides and small molecules is provided in the synthetic methodology section of S1 Text and Figs A and B in S1 Text. Characterization of these compounds are provided in Tables A and B in S1 Text.

## Protein expression and labeling using sortagging

Details describing the expression and purification of Nbs and their labeling using Sortase A are provided in S1 Text. Mass spectrometry characterization of Nbs and Nb conjugates are provided in Table B in S1 Text.

## Statistical analyses

Normalization of data only performed when explicitly mentioned in figure legends. Data with replicates are shown as mean ± SD or mean ± standard error with sample sizes (*n*) listed. For statistical testing, Statistical significance was assessed by one-way ANOVA, with Dunnett's post hoc correction. Statistical comparisons were performed using GraphPad Prism.

## Supporting information

**S1 Text. Supplementary figures, tables, and methods.**
(PDF)

**S1 Data. Raw data used to generate graphics shown in the main text.**
(XLSX)

**S2 Data. Raw data used to generate graphics shown in S1 Text.**
(XLSX)

## Acknowledgments

We thank M. Bouvier (University of Montreal) and J. English (University of Utah) for provision of plasmids used for BRET-based assays of β-Arrestin and G-protein signaling. We acknowledge the intramural mass spectrometry core facility in NIDDK (J. Lloyd) for characterization of peptides and conjugates. We thank S. Ferre (NIH), K. Jacobson (NIH), and F. Ciruela (University of Barcelona) for helpful discussions.

## Author contributions

**Conceptualization:** Shivani Sachdev, Swarnali Roy, Ross W. Cheloha.

**Data curation:** Shivani Sachdev, Swarnali Roy, Ross W. Cheloha.

**Formal analysis:** Shivani Sachdev, Swarnali Roy, Ross W. Cheloha.

**Funding acquisition:** Ross W. Cheloha.

**Investigation:** Shivani Sachdev, Swarnali Roy, Ross W. Cheloha.

**Methodology:** Shivani Sachdev, Swarnali Roy, Ross W. Cheloha.

**Supervision:** Ross W. Cheloha.

**Writing – original draft:** Shivani Sachdev, Swarnali Roy, Ross W. Cheloha.

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
