## [Editor Report · Decision Letter 0]

29 Apr 2025

Dear Dr Cheloha,

Thank you for submitting your manuscript entitled "Nanobody-based conjugates targeting small molecule-binding GPCRs and exhibiting logic-gated signaling" for consideration as a Research Article by PLOS Biology. Please accept my sincere apologies for the delay in getting back to you with feedback.

Your manuscript has now been evaluated by the PLOS Biology editorial staff and I am writing to let you know that we would like to send your submission out for external peer review.

Once your full submission is complete, your paper will undergo a series of checks in preparation for peer review. After your manuscript has passed the checks it will be sent out for review. To provide the metadata for your submission, please Login to Editorial Manager (https://www.editorialmanager.com/pbiology) within two working days, i.e. by May 01 2025 11:59PM.

Kind regards,

Richard

Richard Hodge, PhD

rhodge@plos.org

PLOS

---

## [Decision Letter · Decision Letter 1]

19 Jun 2025

Dear Dr Cheloha,

Thank you for your patience while your manuscript "Nanobody-based conjugates targeting small molecule-binding GPCRs and exhibiting logic-gated signaling" was peer-reviewed at PLOS Biology. Please accept my sincere apologies for the delays that you have experienced during the peer review process. Your manuscript has now been evaluated by the PLOS Biology editors, an Academic Editor with relevant expertise, and by two independent reviewers. Please note that we had recruited a third referee to review the manuscript but they were very late, so we have moved ahead with two reports to avoid any further loss of time.

In light of the reviews, which you will find at the end of this email, we would like to invite you to revise the work to thoroughly address the reviewers' reports.

As you can see, both reviewers are positive about your manuscript and think it is interesting and well done. Reviewer #1 notes that additional data to directly measure receptor-G protein coupling is required, as well as assessing the effects of conjugation on cell surface receptor expression levels for all experiments. In addition, Reviewer #2 notes the revision should include an investigation of biased signaling for the A2a-targeted conjugates.

Given the extent of revision needed, we cannot make a decision about publication until we have seen the revised manuscript and your response to the reviewers' comments. Your revised manuscript is likely to be sent for further evaluation by all or a subset of the reviewers.

**IMPORTANT - SUBMITTING YOUR REVISION**

*Re-submission Checklist*

*Published Peer Review*

*PLOS Data Policy*

*Blot and Gel Data Policy*

Best regards,

Richard

Richard Hodge, PhD

rhodge@plos.org

REVIEWS:

Reviewer #1 (Bryan L Roth, signs review): This is an imaginative and interesting paper that provides a new technology to create modified GPCRs which can function, perhaps, as logic gates. I enjoyed reading this paper and pondering its implications which I think are major.

I have only a couple of technical issues which do not distract from my overall enthusiasm for this work, but which are essential to understand further the potential mechanism(s) involved.

DETAILED AND RELATIVELY MINOR TECHNICAL COMMENTS:

1. FIG 2: did the Nb conjugate alter cell surface receptor expression?

2. The kinetics of activation and inactivation (Figs 2D,F; 3, 4) shoud be replicated with a direct measurement of receptor-G protein coupling (e.g. BRET sensor) to clarify the mechanism (e.g. direct effect on receptor-effector coupling).

3. Also need to assess effects of conjugation on cell surface receptor expression levels for all experiments as it is not entirely clear that the data in Supplementary Fig 6 provides this data for all constructs

Reviewer #2: In this manuscript, Sachdev et al. describe the development of bitopic ligands for the adenosine A2A receptor (A2AR) by using a nanobody against a genetically encoded epitope tag on the extracellular domain of the receptor. The A2A receptor is part of the G protein coupled receptor (GPCR) family which is of broad therapeutic interest. In this work, the authors expand on their prior research, extending the strategy of nanobody-peptide conjugation to a small molecule ligand (CGS21680). This agonist is attached to nanobodies targeting various epitope tags or the extracellular domain of the PTH receptor. In each case, the signaling properties of CGS are altered by its nanobody tethering, in particular showing prolonged activation. Finally, the authors show that using a nanobody targeting one receptor linked to CGS (or a dynorphin peptide) allows "logic-gated" signaling, in which activation of one receptor is contingent on co-expression of another.

Overall, the manuscript is clearly written and interesting. The topic of antibody-drug conjugates is timely and will likely be of interest to a broad readership including researchers in GPCR pharmacology and drug development. The concept of logic-gated activation is clearly demonstrated with several distinct examples provided. While generally thorough, a notable gap in the work is a relatively limited investigation of arrestin activation and signaling bias generally. The authors have shown previously that Nb-peptide conjugates can exhibit strong and unexpected biased signaling, so it would be helpful to see some investigation of this. Specifically, the long duration of action for A2aR-targeted CGS conjugates could in principle reflect differences in arrestin recruitment in addition to direct effects of ligand dissociation kinetics.

Major points:

An investigation (or at a minimum discussion) of biased signaling possibilities for the A2a-targeted conjugates would be appropriate. Some arrestin recruitment assays are provided in the section on using alternative receptors for targeting, but this could be expanded particularly in the context of prolonged receptor activation.

Did the authors examine effects on receptor trafficking when dual-specificity agonists are used? For example, does activation of A2a receptor by CGS force internalization of GLP1R or PTHR? This could be quite important if so, and others have shown that GPCRs can force internalization of other cell surface proteins (Pance et al., Nature Biotech 2022).

The authors describe varying the linker length when developing ligands with agonism of the PTH1R and A2AR, but generally do not comment on initial linker length development of their CGS-Nb. Is this something that did not need to be optimized in this case? If so, it may be worth mentioning.

Minor points:

This paper focuses on presenting functional signaling data of the bitopic ligands developed, which is very common in papers describing antibody drug conjugates but some biochemical characterization of the nanobody ligand conjugate could be interesting to see how biochemical affinities are affected in addition to signaling potency. This would also be important if the authors chose to investigate quantitative analysis of signaling bias.

The click chemistry method to producing nanobody ligand conjugates is of great interest and will be quite useful in future development of bitopic ligands, it would be interesting if the authors stated the percent labeling efficiency of the method.

The structure diagrams in Fig 2 are a bit confusing. The surface representation of the receptor makes it difficult to see that the molecules shown are specifically tethered to the receptor.

---

## [Decision Letter · Decision Letter 2]

3 Sep 2025

Dear Dr Cheloha,

Thank you for your patience while we considered your revised manuscript "Nanobody-based conjugates targeting small molecule-binding GPCRs and exhibiting logic-gated signaling" for publication as a Research Article at PLOS Biology. This revised version of your manuscript has been evaluated by the PLOS Biology editors, the Academic Editor and the original reviewers.

Based on the reviews, I am pleased to say that we are likely to accept this manuscript for publication, provided you satisfactorily address the following data and other policy-related requests that I have provided below (A-F):

(A) We routinely suggest changes to titles to ensure maximum accessibility for a broad, non-specialist readership. In this case, we would suggest a minor edit to the title, as follows. Please ensure you change both the manuscript file and the online submission system, as they need to match for final acceptance:

“Development of bitopic nanobody-ligand conjugates that stimulate G-protein coupled receptor activation and exhibit logic-gated signaling”

(B) You may be aware of the PLOS Data Policy, which requires that all data be made available without restriction: http://journals.plos.org/plosbiology/s/data-availability. For more information, please also see this editorial: http://dx.doi.org/10.1371/journal.pbio.1001797

-Supplementary files (e.g., excel). Please ensure that all data files are uploaded as 'Supporting Information' and are invariably referred to (in the manuscript, figure legends, and the Description field when uploading your files) using the following format verbatim: S1 Data, S2 Data, etc. Multiple panels of a single or even several figures can be included as multiple sheets in one excel file that is saved using exactly the following convention: S1_Data.xlsx (using an underscore).

-Deposition in a publicly available repository. Please also provide the accession code or a reviewer link so that we may view your data before publication.

Figure 2A-F, 3B-C, 4A-G, 5B, 5D-E, S3A, S4B-C, S5B-D, S6A, S7C-E, S8A-D, S9A-B, S10A-B, S11A-B, S12B-C, S13D-F, S14B, S15A-B, S16B-G

(C) Please also ensure that each of the relevant figure legends in your manuscript include information on *WHERE THE UNDERLYING DATA CAN BE FOUND*, and ensure your supplemental data file/s has a legend.

(D) Per journal policy, if you have generated any custom code during the course of this investigation, please make it available without restrictions. Please ensure that the code is sufficiently well documented and reusable, and that your Data Statement in the Editorial Manager submission system accurately describes where your code can be found.

(E) Please note that per journal policy, the model system/species studied should be clearly stated in the abstract of your manuscript.

(F) Please ensure that you are using best practice for statistical reporting and data presentation. These are our guidelines https://journals.plos.org/plosbiology/s/best-practices-in-research-reporting#loc-statistical-reporting and a useful resource on data presentation https://journals.plos.org/plosbiology/article?id=10.1371/journal.pbio.1002128

- If you are reporting experiments where n ≤ 5, please plot each individual data point.

We expect to receive your revised manuscript within two weeks.

*Published Peer Review History*

*Press*

Best regards,

Richard

Richard Hodge, PhD

rhodge@plos.org

Reviewer remarks:

Reviewer #1 (Bryan Roth, identifies himself): My concerns are addressed

Reviewer #2: The authors have very thoroughly addressed all concerns I had about the manuscript, including with the addition of new data. I am supportive of publication.

---

## [Editor Report · Decision Letter 3]

18 Sep 2025

Dear Dr Cheloha,

On behalf of my colleagues and the Academic Editor, Yan Zhang, I am pleased to say that we can accept your manuscript for publication, provided you address any remaining formatting and reporting issues. These will be detailed in an email you should receive within 2-3 business days from our colleagues in the journal operations team; no action is required from you until then. Please note that we will not be able to formally accept your manuscript and schedule it for publication until you have completed any requested changes.

PRESS

Best wishes, 

Richard

Richard Hodge, PhD

rhodge@plos.org

PLOS
